# Study of the Fermentation Characteristics of Non-Conventional Yeast Strains in Sweet Dough

**DOI:** 10.3390/foods12040830

**Published:** 2023-02-15

**Authors:** Evelyne Timmermans, Ine Langie, An Bautil, Kristof Brijs, Carolien Buvé, Ann Van Loey, Ilse Scheirlinck, Roel Van der Meulen, Christophe M. Courtin

**Affiliations:** 1Laboratory of Food Chemistry and Biochemistry, Leuven Food Science and Nutrition Research Centre (LFoRCe), Katholieke Universiteit Leuven, Kasteelpark Arenberg 20, 3001 Leuven, Belgium; 2Laboratory of Food and Microbial Technology, Leuven Food Science and Nutrition Research Centre (LFoRCe), Katholieke Universiteit Leuven, Kasteelpark Arenberg 22, 3001 Leuven, Belgium; 3Vandemoortele Izegem NV, Prins Albertlaan 12, 8870 Izegem, Belgium

**Keywords:** non-conventional yeast strains, volatile compounds, fermentation, yeast metabolites

## Abstract

Despite the diverse functions of yeast, only a relatively homogenous group of *Saccharomyces cerevisiae* yeasts is used in the baking industry. Much of the potential of the natural diversity of yeasts has not been explored, and the sensory complexity of fermented baked foods is limited. While research on non-conventional yeast strains in bread making is increasing, it is minimal for sweet fermented bakery products. In this study, the fermentation characteristics of 23 yeasts from the bakery, beer, wine, and spirits industries were investigated in sweet dough (14% added sucrose *w*/*w* dm flour). Significant differences in invertase activity, sugar consumption (0.78–5.25% *w*/*w* dm flour), and metabolite (0.33–3.01% CO_2_; 0.20–1.26% ethanol; 0.17–0.80% glycerol; 0.09–0.29% organic acids) and volatile compound production were observed. A strong positive correlation (R^2^ = 0.76, *p* < 0.001) between sugar consumption and metabolite production was measured. Several non-conventional yeast strains produced more positive aroma compounds and fewer off-flavors than the reference baker’s yeast. This study shows the potential of non-conventional yeast strains in sweet dough.

## 1. Introduction

Fermented and baked foods are widely consumed in our society. For many years, the yeast *Saccharomyces cerevisiae* has been added to these products [1,2] because this yeast species can ferment efficiently in high-sugar media, uses maltose, shows high CO_2_ production, and has a high ethanol production and tolerance [3,4,5,6]. Moreover, *S. cerevisiae* is widely available and was given the GRAS (Generally Regarded As Safe) status by the Food and Drug Administration (FDA) and the QPS (Qualified Presumption of Safety) status by the European Food Safety Authority (EFSA) because of its history of safe use and absence of toxin production [5]. For many years, the dough-leavening effect resulting from CO_2_ and ethanol production was considered the only important function of yeast.

In recent years, however, there has been more awareness that yeast is not only responsible for dough leavening but also for the production of many aromatic secondary metabolites, such as esters, aldehydes, and ketones that significantly contribute to the flavor profile of bread [7,8,9]. Although aromas can also result from the effects of Maillard reactions, which take place between amino acids and sugars during baking [10,11], and from oxidation of lipids [12], yeast metabolism is reported to be the main source of aromatic diversity in bread [7]. Besides its impact on the organoleptic quality of bread, several components released from yeast cells, such as glycerol, ethanol, organic acids, and glutathione, influence dough characteristics, such as dough strength, elasticity, and extensibility [13,14,15,16,17,18,19]. Despite these diverse functions of yeast in fermented bakery products, only a relatively homogenous group of *S. cerevisiae* yeast strains is widely used for dough leavening. Most of the natural diversity of non-conventional yeasts has not been explored yet for aroma production. As a result, the sensory complexity of fermented and baked foods is limited [3,4]. The increasing demand for fermented bakery products with novel natural aromas justifies the search for non-conventional baker’s yeasts [5]. As a result, research on non-conventional yeast strains in bread has recently increased [3,20,21,22,23].

Hagman et al. [24] already reported that several non-conventional yeast strains exhibit the ability to ferment regardless of the presence of O_2_, also known as the Crabtree effect, meaning that there is a large dough leavening potential among non-conventional yeasts. Research on the sensory aspects of non-conventional yeasts in dough has revealed that several non-conventional yeasts produce different volatile compounds, resulting in more complex flavor profiles in fermented bakery products. Aslankoohi et al. [3], for example, reported *Torulaspora delbrueckii* and *S. bayanus* as candidate leavening agents. Bread produced with these yeast strains was preferred by a trained sensory panel over the conventional baker’s yeast. The bread had a more complex, nutty, forest-like, and aromatic and fruity flavor for bread prepared with *T. delbrueckii* strain Y273 and *S. bayanus* strain Y156, respectively. Zhou et al. [25] observed that using *Kazachstania gamospora* and *Wickerhamomyces subpelliculosus* in bread also resulted in a more complex aroma profile and better overall outcomes in sensory analyses. In addition, these strains showed higher stress tolerance to sugar and salt. Not only does their ability to produce a diverse aroma profile in baked goods make non-conventional yeasts of possible commercial interest, but also their higher tolerance to environmental stresses [26,27]. These so-called “baking-associated stresses” include osmotic stress, thermal stress, salinity stress, oxidative stress, air-drying stress, freezing and thawing stress, ethanol stress, and others [27,28,29]. Despite the huge potential of non-conventional yeasts to increase and diversify the flavor complexity in baked products and their robust stress tolerance, research on the fermentation characteristics of non-conventional yeast strains in sweet dough remains unexplored.

In contrast to the majority of previous studies on non-conventional yeast strains, in which yeasts were tested in lean bread dough [3,20,21,25,30], this study aims to explore the fermentation characteristics of non-conventional yeast strains from different food industries in sweet dough. To this end, the fermentation characteristics and volatile compound production of 23 commercially available yeast strains from the bakery, beer, wine, and spirits industry will be studied in sweet dough with 14% added sucrose (*w*/*w* dm flour). The insights gained on the fermentation characteristics of non-conventional yeast strains in sweet dough could offer opportunities to improve the end product quality of (sweet) fermented bakery products.

## 2. Materials and Methods

### 2.1. Materials

Commercial wheat flour (protein content of 12.5 ± 0.0%, (dry matter (dm) base); moisture content of 13.7 ± 0.1%) was obtained from Paniflower (Merksem, Belgium). The protein level (N × 5.7) was determined in duplicate according to an adaptation of the AOAC method 990.03 with an automated Dumas protein analysis system (VarioMax Cube N, Elementar, Hanau, Germany). The moisture content was analyzed using AACC International Approved Method 44–15.02.23. Fine sucrose was obtained from Tiense Suikerraffinaderij (Tienen, Belgium), and sodium chloride from Everyday (Halle, Belgium). A total of 23 yeast strains were investigated in this study. The industry, species, and origin of these yeast strains are presented in Table 1. All reagents, solvents, and chemicals were of analytical grade and obtained from Sigma-Aldrich (Bornem, Belgium) unless indicated otherwise.

### 2.2. CO_2_ Production Analysis

To measure CO_2_ production in yeasted dough, dough was prepared by mixing 10.0 g flour, 5.4 mL deionized water, 1.2 g sucrose, 0.3 g dry yeast, and 0.2 g salt for 4 min with a 10 g pin bowl mixer (National Manufacturing, Lincoln, NE, USA). Next, the dough samples were fermented in Risograph canisters (400 mL, National Manufacturing, Lincoln, NE, USA) in a water bath at 30 °C for 120 min. The CO_2_ production was measured every minute automatically with a Risograph system. Measurements were performed in triplicate, starting from three biological replicates.

### 2.3. Quantification of Mono-, Di- and Trisaccharides

To analyze the sugar content in dough with non-conventional yeast strains, dough samples were prepared and fermented as described in paragraph 2.2. After 120 min fermentation (30 °C), the samples were frozen in liquid nitrogen, lyophilized and ground with a mortar and pestle to obtain a powder with a moisture content of 3–4%. Next, enzymes were inactivated by adding an 80% (*v*/*v*) ethanol solution, and rhamnose (8.0 mg/mL) was added as an internal standard to quantify the saccharides. Saccharides were extracted with water and separated on a Dionex ICS5000 chromatography system (Thermo Fisher Scientific, Waltham, MA, USA), as previously described by Laurent et al. [31]. Measurements were performed in triplicate, starting from three biological replicates. Saccharide concentrations are expressed as weight percentages on flour dm base (% *w*/*w* dm flour). The glucose and fructose consumption by yeast was estimated by calculating the difference between the initial glucose and fructose concentrations coming from the flour and from sucrose and fructan hydrolysis [32], and the final glucose and fructose concentrations. However, this calculation is an estimation since the amount of glucose and fructose from fructan hydrolysis was estimated based on previous measurements on the reference dough [32].

### 2.4. Quantification of Ethanol, Glycerol and Organic Acids

To quantify ethanol, glycerol and organic acids produced by non-conventional yeast strains, fermented dough samples were prepared as described in paragraph 2.2. After a fermentation time of 120 min (30 °C), extraction was performed by blending the dough samples (15.0 g) with deionized water (two times the amount of the dough sample) for 30 s with a Waring 8011E blender (Waring Products, Torrington, CT, USA) [22]. The extracts were further prepared and analyzed with Ion-Exclusion High-Performance Liquid Chromatography. The HPLC system (Shimadzu, Kyoto, Japan) consisted of an LC-20AT pump, a DGU-20A5 degasser, a SIL-HTc autosampler, a CTO-20A column oven and a Refractive Index Detector 10A. The conditions used to separate ethanol, glycerol and organic acids were identical to those described previously by Timmermans et al. [32]. A Rezex ROA-Organic acid ion-exclusion analytical column (with guard) was used at 60 °C with 2.5 mM H_2_SO_4_ solution as mobile phase and a flow rate of 0.6 mL/min. Measurements were performed in triplicate, starting from three biological replicates. Ethanol, glycerol, and organic acid concentrations are expressed as weight percentages on flour dm base (% *w*/*w* dm flour).

### 2.5. Analysis of Volatile Compounds

The volatile compounds in fermented sweet dough with the reference baker’s yeast and non-conventional yeast strains were analyzed according to a method adapted from Aslankoohi et al. [3]. The aim was to screen the volatile fraction in the doughs without focusing on specific compounds. A headspace solid-phase microextraction (HS-SPME) was followed by gas chromatography-mass spectrometry (GC–MS) analysis, with a GC system 7890B, coupled to an MS detector 5977A (Agilent Technologies, Santa Clara, CA, USA) and an autosampler (PAL system, CTC Analytics AG, Zwinge, Switzerland). For each sample, 5.00 g of dough was put in a 20-mL glass vial, sealed with a silicone septum, and kept in −80 °C until analysis. Right before analysis, the samples were thawed at room temperature and incubated in the autosampler of the GC-MS system at 40 °C under agitation (500 rpm) for 5 min. After this equilibration period, a 25/120 μm DVB/CAR/PDMS fiber (Supelco, Bellefonte, PA, USA) was introduced into the headspace of the vial by piercing the septum of the cap. Subsequently, an extraction took place under agitation (30 min, 40 °C). Next, the fiber was introduced into the injection port of the GC system where it was heated for 300 s at 130 °C to desorb the volatile compounds. The sample was injected in splitless mode on a polar column (HP Innowax, 60 m × 250 μm × 0.25 μm) with helium as carrier gas (flow rate of 1.14 mL/min, 124.88 kPa). The oven temperature was first increased from 40 °C to 100 °C at a heating rate of 4 °C/min, and was then increased further to 114 °C at 1 °C/min. After a holding time of 6 min at 114 °C, the temperature was increased to 136 °C at 6 °C/min. Lastly, the temperature was increased at a heating rate of 85 °C to a final temperature of 245 °C, which was held for 2 min. The mass detector operated in scan mode at 230 °C (20–400 amu), using electron impact ionization (70 eV). A mix of linear *n* alkanes (from C_8_ to C_19_) was analyzed under identical conditions to serve as external retention index markers. Chromatograms were deconvoluted to obtain pure compound spectra using an Automated Mass Spectral Deconvolution and Identification System (AMDIS) software (Version 2.72, 2014, National Institute of Standards and Technology, Gaithersburg, MD, USA) and matched to a commercial GC/MS library (NIST14 database). Finally, Mass Profiler Professional software (MPP) (version B12.00, 2012, Agilent Technologies, Diegem, Belgium) was used for baseline correction and peak alignment and filtering, resulting in a data table with all peak areas. For each yeast strain, three biological replicates were analyzed.

### 2.6. Statistical Analysis

Statistical evaluation of the results was performed using the Fit Model platform of the statistical software JMP Pro 15 (SAS Inst., Cary, NC, USA). Significant differences in sugar or metabolite content between yeast strains were determined by one-way analysis of variance (ANOVA) with comparison of mean values using the posthoc Tukey HSD test (α = 0.05). Multivariate correlation analysis was used to obtain the coefficients of determination, *R^2^*, and the correlation probabilities *p* to study the correlation between sugar consumption and metabolite production. To test for differences in the volatile compounds in dough samples prepared with the different yeast strains, the log2-transformed relative peak areas (the log2 of the peak area divided by the total peak areas of all detected compounds) were statistically compared using linear mixed models in which biological repeat was used as a random effect and yeast as a fixed effect. The posthoc Tukey HSD test was used to identify significantly different means between the yeast strains. In addition, a Dunnett test was used to identify significant differences between the reference baker’s yeast and the non-conventional yeast strains for each volatile compound.

## 3. Results

To detect differences in fermentation characteristics in sweet dough between 23 different yeast strains, the concentrations of sugars (glucose, fructose, sucrose, and maltose) and secondary yeast metabolites (ethanol, glycerol, succinic acid, and acetic acid), the CO_2_ production, and the relative concentrations of volatile compounds in dough were measured after a fermentation step of 120 min at 30 °C.

### 3.1. Sugar Consumption

In all doughs, except for the doughs with WB06, BE134, and BCool, no sucrose was measured after 120 min of fermentation due to invertase activity (Figure 1).

Glucose and fructose concentrations ranged from 4.05% to 6.90% and from 6.40 to 7.81%, respectively, in these doughs. In the doughs with WB06, BE134, and BCool, sucrose was still present in relatively high concentrations (3.30–5.10%). As a result, glucose and fructose concentrations were lower compared to the other doughs, ranging from 2.96% to 4.22%, and from 3.64% to 4.56%, respectively. In all doughs, maltose was measured in concentrations of 1.01% to 2.01%. Based on the glucose and fructose concentrations, the amount of glucose and fructose consumption was calculated (Figure 2).

Differences in sugar consumption by different yeast strains were observed (*p* < 0.001). SD consumed a significantly higher amount of glucose and fructose compared to the reference yeast (*p* < 0.001), while BDoux and BCool consumed a comparable amount of sugars. NE and NT consumed the lowest amount of sugars.

### 3.2. Metabolite Production

The production of CO_2_, ethanol, glycerol, and organic acids was observed in all (sweet) doughs, although some yeast strains produced a significantly lower amount of fermentation metabolites (Table 2). NE produced the lowest amount of CO_2_ (0.33 ± 0.02%). In dough with NE, also the lowest amount of ethanol (0.20 ± 0.03%), glycerol (0.17 ± 0.01%), and acetic acid (0.02 ± 0.00%) was measured. The highest CO_2_ production was measured in dough with SD (3.01 ± 0.11%). Ethanol, glycerol, succinic acid, and acetic acid were measured in concentrations of 1.26 ± 0.13%, 0.80 ± 0.05%, 0.19 ± 0.01%, and 0.10 ± 0.01%, respectively, in this dough, which was also significantly higher compared to the majority of the other doughs (*p* < 0.001), except for doughs made with T58 and the reference strain. The acetic acid concentration in the Yin sample was significantly higher than in the dough prepared with SD (Table 2).

### 3.3. Relation between Sugar Consumption and Metabolite Production

In Figure 3, the sum of the CO_2_ production and the measured ethanol, glycerol, and organic acid concentrations in the doughs with the different yeast strains are plotted against the calculated sugar consumption amounts.

A strong positive correlation was found with an R^2^ of 0.76 (*p* < 0.001). The different variables were also compared (Table 3). A strong positive correlation was found between CO_2_ and ethanol (R^2^ = 0.95, *p* < 0.001) and between CO_2_ and glycerol (R^2^ = 0.94, *p* < 0.001). Sugar consumption and CO_2_ production were less strongly correlated (R^2^ = 0.74, *p* < 0.001). Only a moderate correlation was found between sugar consumption and succinic acid concentration (R^2^ = 0.44, *p* < 0.001) and between sugar consumption and acetic acid concentration (R^2^ = 0.42, *p* < 0.001).

### 3.4. Volatile Compounds in Sweet Dough

Next to CO_2_, ethanol, glycerol, and organic acids, volatile compounds are also important yeast metabolites because they can impact the flavor profile of the end product [3,7]. A total of 51 different volatile compounds, including 11 aldehydes, 10 esters, 5 carbonic acids, 17 alcohols, 7 ketones, and γ-nonalactone were identified in the 23 fermented doughs (Table 4). A heatmap was made with 23 identified volatile compounds with a known aroma and a significantly different concentration between the reference yeast and the other yeasts for at least one of the volatile compounds shown in this heatmap (Figure 4).

2-Methyl-1-propanal and 3-methylbutanal were found in a significantly higher concentration in dough with NT, NE, BE134, and BCool, compared to the reference dough (*p* < 0.001). 2-Methyl-1-propanal was also found in a significantly higher concentration in dough with Kveik (*p* < 0.001). Ethyl-3-methylbutanoate was not detected in the reference dough, but it was detected in a high concentration in dough with M21. Heptanal and hexanal were found in a significantly higher concentration in dough with NT, NE, BE134, and BCool, compared to the reference dough, whereas 1-heptanol was found in a higher concentration in every dough, except the doughs with BE134, HD54, NDA21, BCool, and Yin (*p* < 0.001). Nonanal was found in a significantly higher concentration in every dough compared to the reference dough, except doughs with CN, Flor, Kveik, R2056, S6U, and W&P (*p* < 0.001). *E*-2-nonenal was only detected in dough with NT, NE, and BCool. By comparing the relative concentrations of the volatile compounds and the CO_2_ production in the different doughs, a positive correlation was found for ethyl acetate (R^2^ = 0.73), isoamyl acetate (R^2^ = 0.76), ethyl hexanoate (R^2^ = 0.81), and ethyl octanoate (R^2^ = 0.76) (*p* < 0.001), whereas a negative correlation was found for 3-pentanone (R^2^ = −0.80, *p* < 0.001) and 1-octen-3-ol (R^2^ = −0.61, *p* = 0.002). Finally, when the role of the yeast origin was investigated, it was found that in the doughs prepared with spirits yeasts, the concentration of alcohols was significantly higher (*p* = 0.001) compared to the other doughs. Isobutyric acid (*p* < 0.015) and butanoic acid (*p* < 0.001) were also present in significantly higher concentrations in doughs with spirits yeasts compared to the other doughs. The concentration of ethyl acetate was significantly higher in doughs with bakery yeasts compared to doughs with beer yeasts (*p* < 0.014). Isobutanol was found in a significantly higher concentration in doughs with bakery yeasts compared to doughs with wine yeasts (*p* < 0.031), while the concentration of isopentanol was significantly higher in doughs with wine or beer yeasts compared to doughs with bakery yeasts (*p* < 0.007).

## 4. Discussion

The purpose of this study was to explore the fermentation characteristics of non-conventional yeast strains from the bakery, beer, wine, and spirits industry in sweet dough with 14% added sucrose (*w*/*w* dm flour). Because differences in fermentation characteristics can influence dough and product characteristics, such as dough rheology and product volume, color and flavor, more insights into the fermentation characteristics can help to implement these yeast strains in bakery products. Moreover, a known aroma profile of different yeast strains can help to select yeast strains for improving product flavor.

To investigate the fermentation characteristics of non-conventional yeast strains in sweet dough, the CO_2_ production and the concentrations of mono-, and disaccharides, glycerol, ethanol, and organic acids were measured in doughs with 23 different yeast strains and 14% added sucrose (*w*/*w* dm flour), after fermentation for 120 min at 30 °C. In all doughs, except the doughs with WB06, BE134, and BCool, sucrose was completely hydrolyzed into glucose and fructose by yeast invertase [37]. In the doughs with WB06, BE134, and Bcool, part of the sucrose was retained in the dough. It can therefore be assumed that these three yeast strains have a lower invertase production or activity compared to the other yeast strains. Laurent et al. [31] also found high variability in the capacity of different yeast strains to hydrolyze sucrose and fructo-oligosaccharides. According to Laurent et al. [38], this variation in invertase activity is caused by the natural variation in SUC gene sequences. Significant differences were also measured in the glucose, fructose, and maltose content between the doughs with non-conventional yeast strains (*p* < 0.001). These differences were caused by variations in consumption dynamics between the yeast strains. The highest sugar consumption was observed in dough with SD. This was to be expected since SD is commercially available as an osmotolerant baker’s yeast. The lowest sugar consumption was observed in the doughs with NE and NT, which are, respectively, a beer and a spirits yeast. This can probably be explained by the short fermentation time applied in the current study. In the production process of beer, wine, and spirits, the fermentation time is typically much longer compared to the production of bakery products. It is possible that the fermentation rate of beer, wine and spirits yeasts will be higher when fermentation times are longer [39]. Although variation in invertase activity was observed, this difference did not necessarily lead to a lower sugar consumption compared to the reference yeast. BCool, for example, had a reduced invertase activity but still a relatively high sugar consumption compared to the other non-conventional yeast strains. We can therefore assume that invertase activity does not influence the fermentation characteristics of yeast. Due to the highly variable sugar concentrations in doughs (2.96–7.12%, 3.64–7.81%, and 0.00–5.10% for glucose, fructose, and sucrose, respectively) among the different yeast strains, it can be assumed that the use of non-conventional yeast strains will also impact product characteristics, such as product color and sweetness [40].

When the sum of the CO_2_ production and the measured ethanol, glycerol, and organic acid concentrations in the doughs with the different yeast strains was plotted against the calculated sugar consumption amounts, a strongly positive correlation was found (R^2^ = 0.76) (Figure 3). This was expected, as yeast converts the sugars during fermentation into CO_2_, ethanol, glycerol, and other secondary metabolites [28]. However, it was not a perfect fit because not all fermentation products were taken into account (e.g., glutathione, volatile compounds), and the sugar consumption is only an approximate calculation. When the different variables were compared, the strongest correlation was found between CO_2_ and ethanol (R^2^ = 0.95) and between CO_2_ and glycerol (R^2^ = 0.94). This is because CO_2_, ethanol, and glycerol are all produced and released during glycolysis and alcoholic fermentation [41]. Sugar consumption and succinic acid (R^2^ = 0.44) and sugar consumption and acetic acid (R^2^ = 0.42) were only moderately correlated, which can be explained by the fact that both organic acids are partly used in the citric acid cycle and, hence, not completely released after production in the yeast cell [41].

As we were also interested in the differences in the aroma profile among different yeast strains, the last group of yeast metabolites that were analyzed in the different doughs were volatile compounds. Research on the volatile compounds present in bread and lean bread dough is abundant [7,8,9,11,12,33,42,43]. Volatile compounds identified in fermented bread crumb are mainly derived from yeast metabolism and from lipid oxidation [7,9,43], whereas the volatile compounds in the crust arise from Maillard reactions [10]. In this study, we only analyzed the volatile compounds in sweet fermented dough, as the focus of this study was oriented toward the further investigation of the influence of yeast fermentation on the volatile compounds and not on the influence of the baking step. In the literature, no descriptions of aroma profiles in sweet dough or fermented pastry products were found. For this reason, aroma profiles of bread were used to interpret our results [3,7,8,12,33,35]. However, it is not always easy to associate a volatile compound with only a pleasant flavor or only an off-flavor, as some off-flavors can act as pleasant flavors in low concentrations. Acetic acid might, for example, act as an aroma enhancer when present in low concentrations in wheat bread crumb [7].

The majority of the detected compounds in sweet dough were also found in volatile compound analyses on bread [3,7,8,9,12,33,44], although some compounds, such as 2-methyl-1-propanal, *E*-2-heptenal, ethyl isobutyrate, isopentanol, 3-pentanone, and 3-octanone, were not detected in these studies. Conversely, some other compounds were found in the studies on bread that were not found in the current study. Probably, most of these differences in the presence or absence of volatile compounds are due to differences in GC-MS method and not in yeast strain, as the above-mentioned compounds were measured in most of the doughs with different yeast strains in the current study. However, it is also possible that the higher sugar content or the dough-making process had an influence on the aroma production. In addition to the method used, yeast strain and dough specifications, the baking process should also be taken into account when comparing detected volatile compounds among different studies. Volatiles in dough and the baked product might largely differ as the high temperatures during the baking step can, for instance, lead to a loss of (pleasant) aromas and, as mentioned previously, other biochemical reactions will occur during baking.

Birch et al. [8] and Frasse et al. [9] found that most of the aroma compounds in the crumb of fermented bread are derived from the metabolism of yeast and that the dominating compounds are alcohols, aldehydes as well as 2,3-butanedione (diacetyl), 3-hydroxy-2-butanone (acetoin) and esters. It can be assumed that yeast strains with a high fermentation rate (i.e., high CO_2_ production) result in higher concentrations of yeast metabolism-derived compounds. Indeed, in the current study, a strong positive correlation was found for the esters ethyl acetate (R^2^ = 0.73), isoamyl acetate (R^2^ = 0.76), ethyl hexanoate (R^2^ = 0.81), and ethyl octanoate (R^2^ = 0.76), which are all derived from the metabolism of yeast [3,7,8]. A strong negative correlation was found for 3-pentanone (R^2^ = −0.81), and a less strong negative correlation for 1-octen-3-ol (R^2^ = −0.61), which are both derived from lipid oxidation [3,44,45]. Esters are often characterized as having a pleasant, fruity, and sweet aroma [7,8,46,47]. It is, therefore, possible that yeast strains with high fermentation rates, like SD or T58, positively influence the aroma of fermented bakery products (Figure 4). High levels of aldehydes and ketones are typically associated with unfermented raw materials or samples at earlier stages of fermentation [3,9]. However, in the present study, a negative correlation with the fermentation rate was only found for one ketone (3-pentanone) and no aldehydes. This might be related to the fact that only one fermentation time was investigated.

The importance of aroma compounds in the food matrix depends not only on their concentration but also on their odor threshold (OT) values. The used OT values were determined in water [8,12,33,34]. Fermented bakery products are, however, complex food matrixes. It would have been more appropriate to use OT values in a starch or cellulose matrix, but these values were not available in the literature for the majority of the compounds. The detected yeast-derived metabolism compounds with the lowest OT value (<1) were 2-methyl-1-propanal, 3-methylbutanal, ethyl-3-methylbutanoate, and 2,3-butanedione, which are all considered to have pleasant aromas in bread [7,11,33,44]. In doughs with NT, NE, BE134, BCool, Kveik, and M21, significantly higher concentrations of these compounds were detected compared to the reference dough (Figure 4). These yeast strains can, hence, be considered interesting yeast strains for aroma production. Only 2,3-butanedione was not found in higher concentrations in doughs with non-conventional yeast strains compared to the reference dough.

After fermentation, the second most important pathway in aroma formation in bakery products is lipid oxidation [8]. Birch et al. [8] found that the level of lipid oxidation products is independent of yeast concentration for the majority of the lipid oxidation products. The detected lipid oxidation compounds with the lowest OT value (< 1) in our doughs were hexanal, heptanal, nonanal, *E*-2-nonenal, and 1-heptanol. According to Aslankoohi et al. [3], 1-heptanol and heptanal did not vary across bread prepared with different *S. cerevisiae* strains. However, we found significantly different concentrations of heptanal in the different doughs, often higher compared to the reference dough. Hexanal, 1-heptanol, and nonanal were also found in significantly higher concentrations in several doughs compared to the reference dough. *E*-2-nonenal was only detected in dough with NT, NE, and BCool. It can therefore be assumed that yeast has an indirect effect on the formation of some lipid oxidation compounds. Birch et al. [8] also observed that the formation of 2-octanone and octanal significantly increased with increasing yeast concentration, although they concluded that the formation of most lipid oxidation products is not related to yeast metabolism. Lipid oxidation compounds are often characterized as being off-flavors [7]. Therefore, doughs with low levels of these compounds might result in higher consumer acceptance. In the case of nonanal and *E*-2-nonenal, there have been some controversies [12]. On the one hand, they have been described as odorants with pleasant aroma properties due to their citrus and cucumber notes, but on the other hand, they have also been characterized by fatty-tallow notes [12]. This probably depends on the concentration and presence of other volatiles. Three doughs contained significantly less lipid oxidation-derived compounds (*p* < 0.001), namely doughs with Flor, S6U, and R2056 (Figure 4). Therefore, these yeast strains could possibly result in a higher consumer acceptance of the bakery product.

Next, the impact of the origin of the yeast strain on the aroma profile was investigated. There were no significant differences found in the level of aldehydes, esters, acids, and ketones between the different yeast groups, but in the doughs prepared with spirits yeasts, the concentration of alcohols was significantly higher (*p* < 0.001) compared to the other doughs (Figure 4). This was to be expected, as these yeast strains are used to produce high alcohol-containing beverages. When they are used in the production of bakery products, this can result in a more alcoholic flavor.

In the future, it would be interesting to quantify some aroma-active compounds that were present in relatively different amounts in the doughs in order to investigate if they exceed the OT value. It would also be useful to prepare baked fermented pastry products with a selection of non-conventional yeast strains, as the baking step can cause evaporation of aroma compounds with a low boiling point but can also result in the formation of new aroma compounds. In contrast to bread, the high fat concentration in pastry might result in better retention of fat-soluble aroma compounds during baking [48]. Moreover, fat can provide precursors for the formation of new aroma compounds [48]. Next, sensory analyses should be performed to investigate if consumers can notice the changed aroma profile and if there is a difference in preference.

## 5. Conclusions

This study aimed to investigate the fermentation characteristics of non-conventional yeast strains from different food industries in sweet dough with 14% added sucrose (*w*/*w* dm flour). The 22 non-conventional yeast strains were able to ferment in sweet dough, although at different fermentation rates, which resulted in diverse sugar and metabolite concentrations in the doughs. A significant strong positive correlation (R^2^ = 0.76) was observed between sugar consumption and CO_2_, ethanol, glycerol, and organic acid production. In addition, several yeast strains produced more positive aroma compounds compared to reference baker’s yeast, which is desired for the production of fermented bakery products. The results in this work contribute to the exploration of fermentation characteristics of non-conventional yeast strains in sweet dough, which is justified by the increasing demand for fermented bakery products with novel natural aromas. In the future, it would be interesting to prepare fermented pastry products with selected non-conventional yeast strains to investigate the impact of the baking phase and the presence of a high fat content on the end product aroma profile. Sensory analysis should be performed to investigate consumer liking and acceptance.

## Figures and Tables

**Figure 1 foods-12-00830-f001:**
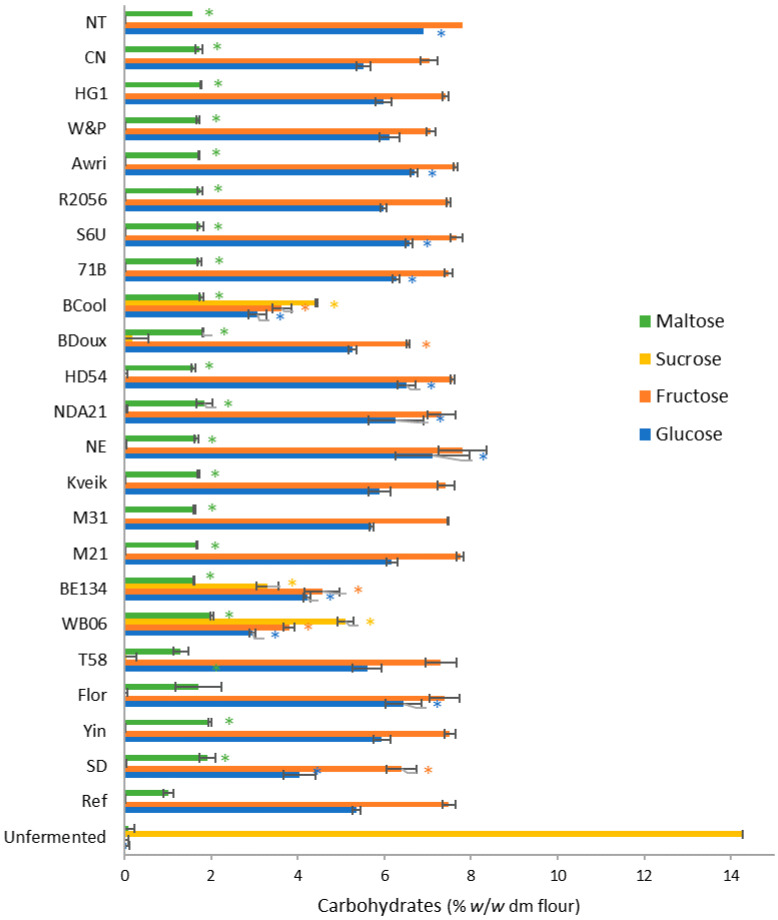
Glucose, fructose, sucrose, and maltose concentrations in sweet doughs (14% sugar, *w*/*w* dm flour) made with 23 different yeast strains after a fermentation step of 120 min at 30 °C. Concentrations are expressed on flour dry matter base (% *w*/*w* dm flour). The standard deviations result from triplicate measurements. ‘*’ indicates concentrations that were significantly different from the reference (ref) (*p* < 0.05).

**Figure 2 foods-12-00830-f002:**
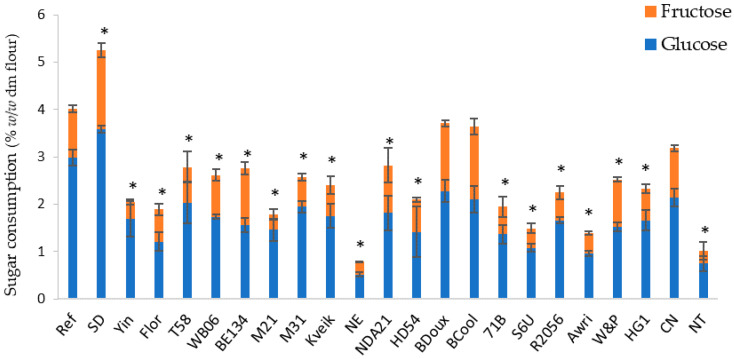
Glucose and fructose consumption levels in sweet doughs (14% sugar, *w*/*w* dm flour) with 23 different yeast strains after a fermentation step of 120 min at 30 °C. Concentrations are expressed on flour dry matter base (% *w*/*w* dm flour). The standard deviations result from triplicate measurements. ‘*’ indicates concentrations that were significantly different from the reference (*p* < 0.05).

**Figure 3 foods-12-00830-f003:**
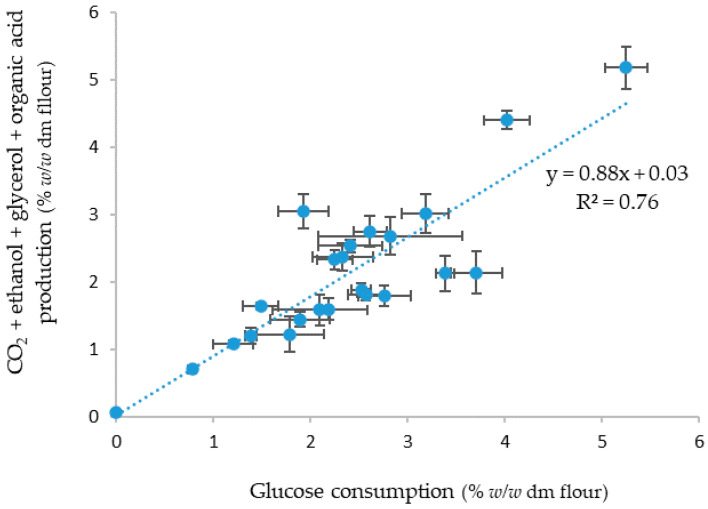
Correlation between the sum of the CO_2_ production and the measured ethanol, glycerol, and organic acid concentrations in sweet doughs (14% sugar, *w*/*w* dm flour). Vertical bars represent standard deviations of triplicate metabolite measurements and horizontal bars represent standard deviations of triplicate consumption measurements.

**Figure 4 foods-12-00830-f004:**
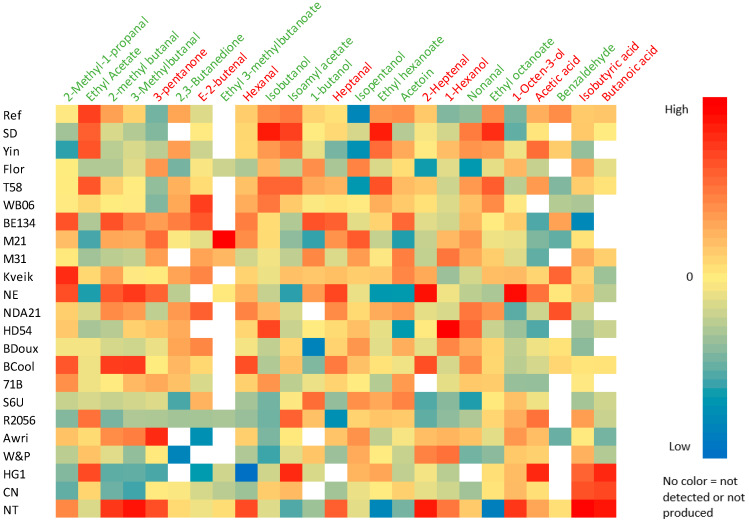
Heatmap illustrating the differences in relative concentrations of volatile compounds, as indicated by contrasting color codes (listed on the right), in sweet dough (14% sugar, *w*/*w* dm flour) prepared with 23 different yeast strains. Data are based on HS-SPME-GC-MS analysis of three biological replicates. For each column, the values shown in the heatmap are z-scores calculated from the average log2 relative peak areas (i.e., how many standard deviations any given value deviates from the mean of the column) and show if a certain strain produced a lower (blue tainted boxes) or higher (red tainted boxes) amount of a certain volatile compared to the mean value of that volatile compound. The compounds in green text are considered compounds with a positive aroma in bread, while the compounds in red text are considered off-flavors in bread.

**Table 1 foods-12-00830-t001:** Reference baker’s yeast and 22 non-conventional yeast strains from the bakery, wine, beer, or spirits industry. Species and origins are shown.

Name	Code	Industry	Species	Origin
Instant (reference)	Ref	Bakery	*S. cerevisiae*	Algist Bruggeman (Gent, Belgium)
Instant SD	SD	Bakery	*S. cerevisiae*	Algist Bruggeman (Gent, Belgium)
Yeast-in	Yin	Bakery	*S. cerevisiae*	Algist Bruggeman (Gent, Belgium)
Florapan^®^ Aromatic Yeast	Flor	Bakery	*S. cerevisiae*	Lallemand (Montréal, QC, Canada)
SafAle T-58	T58	Beer	*S. cerevisiae*	Fermentis (Marcq-en-Barœul, France)
SafAle WB-06	WB06	Beer	*S. cerevisiae* var. *diastaticus*	Fermentis (Marcq-en-Barœul, France)
LalBrew^®^ New England	NE	Beer	*S. cerevisiae*	Lallemand (Montréal, QC, Canada)
Belgian Wit M21	M21	Beer	*S. cerevisiae*	Mangrove Jack’s (Auckland, New Zealand)
Belgian Tripel M31	M31	Beer	*S. cerevisiae*	Mangrove Jack’s (Auckland, New Zealand)
SafAle BE-134	BE134	Beer	*S. cerevisiae* var. *diastaticus*	Fermentis (Marcq-en-Barœul, France)
LalBrew^®^ Voss Kveik Ale	Kveik	Beer	*S. cerevisiae*	Lallemand (Montréal, QC, Canada)
SafŒno NDA 21	NDA21	Wine	*S. cerevisiae*	Fermentis (Marcq-en-Barœul, France)
SafŒno HD54	HD54	Wine	*S. cerevisiae x S. cerevisiae* var. *bayanus*	Fermentis (Marcq-en-Barœul, France)
Awri ZEVII	Awri	Wine	*S. cerevisiae x S. kudriavzevii*	Maurivin (Hampton, UK)
Bioferm Doux	BDoux	Wine	*S. cerevisiae* var. *bayanus*	Vinoferm (Beverlo, Belgium)
Oeneferm Wild and Pure	W&P	Wine	*Torulaspora delbrückii*	Erbslöh (Geisenheim, Germany)
Lalvin^®^ 71B	71B	Wine	*S. cerevisiae*	Lallemand (Montréal, QC, Canada)
Lalvin^®^ Rhône 2056	R2056	Wine	*S. cerevisiae*	Lallemand (Montréal, QC, Canada)
Bioferm Cool	BCool	Wine	*S. cerevisiae* var. *bayanus*	Vinoferm (Beverlo, Belgium)
Lalvin^®^ S6U	S6U	Wine	*S. cerevisiae* var. *uvarum*	Lallemand (Montréal, QC, Canada)
SafSpirit HG-1	HG1	Spirits	*S. cerevisiae*	Fermentis (Marcq-en-Barœul, France)
Distillamax CN	CN	Spirits	*S. cerevisiae*	Lallemand (Montréal, QC, Canada)
Distillamax NT	NT	Spirits	*S. cerevisiae*	Lallemand (Montréal, QC, Canada)

**Table 2 foods-12-00830-t002:** CO_2_ production and ethanol, glycerol, succinic acid, and acetic acid concentrations in doughs with 23 different yeast strains after a fermentation step of 120 min at 30 °C. Concentrations are expressed on flour dry matter base (% *w*/*w* dm flour). Standard deviations of triplicate measurements are presented for each product.

Sample	CO_2_ (% *w*/*w* dm flour) ^1^	Succinic Acid (% *w*/*w* dm flour) ^1^	Glycerol (% *w*/*w* dm flour) ^1^	Acetic Acid (% *w*/*w* dm flour) ^1^	Ethanol (% *w*/*w* dm flour) ^1^
Initial dough	0.00 ± 0.00	0.02 ± 0.00	0.03 ± 0.01	0.01 ± 0.00	0.00 ± 0.00
Ref	2.61 ± 0.16 ^b^	0.16 ± 0.02 ^abc^	0.73 ± 0.07 ^a^	0.12 ± 0.02 ^ab^	0.98 ± 0.17 ^bc^
SD	3.01 ± 0.11 ^a^	0.19 ± 0.01 ^a^	0.80 ± 0.05 ^a^	0.10 ± 0.01 ^b^	1.26 ± 0.13 ^a^
Yin	1.70 ± 0.09 ^c^	0.16 ± 0.00 ^bc^	0.59 ± 0.04 ^b^	0.13 ± 0.01 ^a^	0.62 ± 0.11 ^def^
Flor	0.78 ± 0.03 ^jklm^	0.07 ± 0.01 ^gh^	0.29 ± 0.02 ^ghi^	0.04 ± 0.01 ^defgh^	0.37 ± 0.05 ^fghi^
T58	2.45 ± 0.04 ^b^	0.17 ± 0.01 ^ab^	0.78 ± 0.03 ^a^	0.11 ± 0.02 ^ab^	1.17 ± 0.18 ^ab^
WB06	1.64 ± 0.16 ^cd^	0.14 ± 0.01 ^cd^	0.40 ± 0.03 ^cdef^	0.06 ± 0.00 ^cd^	0.66 ± 0.03 ^de^
BE134	1.03 ± 0.11 ^ghij^	0.07 ± 0.01 ^gh^	0.31 ± 0.01 ^efghi^	0.04 ± 0.00 ^defgh^	0.44 ± 0.02 ^efghi^
M21	0.66 ± 0.13 ^klm^	0.08 ± 0.01 ^fgh^	0.24 ± 0.04 ^ij^	0.03 ± 0.00 ^efgh^	0.32 ± 0.09 ^ghi^
M31	0.99 ± 0.04 ^hij^	0.08 ± 0.00 ^fgh^	0.33 ± 0.01 ^efghi^	0.03 ± 0.00 ^fgh^	0.49 ± 0.04 ^efgh^
Kveik	1.38 ± 0.02 ^def^	0.09 ± 0.00 ^fg^	0.49 ± 0.04 ^bc^	0.05 ± 0.00 ^cde^	0.62 ± 0.03 ^def^
NE	0.33 ± 0.02 ^n^	0.07 ± 0.00 ^gh^	0.17 ± 0.01 ^j^	0.02 ± 0.00 ^h^	0.20 ± 0.03 ^i^
NDA21	1.64 ± 0.09 ^cd^	0.12 ± 0.02 ^de^	0.45 ± 0.08 ^cd^	0.06 ± 0.01 ^cd^	0.55 ± 0.08 ^defgh^
HD54	0.85 ± 0.10 ^ijkl^	0.15 ± 0.00 ^bc^	0.29 ± 0.02 ^ghi^	0.05 ± 0.01 ^cdefg^	0.39 ± 0.09 ^fghi^
BDoux	1.21 ± 0.16 ^efgh^	0.08 ± 0.01 ^fgh^	0.36 ± 0.03 ^defgh^	0.05 ± 0.00 ^defgh^	0.55 ±0.11 ^defgh^
BCool	1.18 ± 0.13 ^efgh^	0.09 ±0.01 ^fg^	0.34 ± 0.03 ^efghi^	0.05 ± 0.01 ^cdefgh^	0.56 ± 0.10 ^defg^
71B	0.86 ± 0.08 ^ijkl^	0.07 ± 0.00 ^gh^	0.32 ± 0.03 ^efghi^	0.04 ± 0.00 ^defgh^	0.39 ± 0.04 ^fghi^
S6U	0.88 ± 0.01 ^ijk^	0.08 ± 0.01 ^fgh^	0.29 ± 0.00 ^fghi^	0.03 ± 0.01 ^gh^	0.46 ± 0.03 ^efghi^
R2056	1.30 ± 0.07 ^efg^	0.08 ± 0.01 ^fgh^	0.42 ± 0.02 ^cde^	0.07 ± 0.01 ^c^	0.56 ± 0.04 ^defg^
Awri	0.59 ± 0.05 ^lmn^	0.08 ± 0.01 ^fgh^	0.30 ± 0.03 ^fghi^	0.05 ± 0.00 ^cdefgh^	0.29 ± 0.03 ^hi^
W&P	1.09 ± 0.06 ^fghi^	0.06 ± 0.01 ^h^	0.27 ± 0.02 ^hij^	0.03 ± 0.01 ^efgh^	0.53 ± 0.02 ^defgh^
HG1	1.38 ± 0.11 ^de^	0.07 ± 0.01 ^gh^	0.39 ± 0.03 ^cdefg^	0.05 ± 0.00 ^cdef^	0.60 ± 0.05 ^def^
CN	1.62 ± 0.08 ^cd^	0.10 ± 0.01 ^ef^	0.57 ± 0.06 ^b^	0.05 ± 0.01 ^cdefgh^	0.78 ± 0.14 ^cd^
NT	0.50 ± 0.03 ^mn^	0.07 ± 0.00 ^gh^	0.26 ± 0.01 ^hij^	0.04 ± 0.01 ^defgh^	0.30 ± 0.01 ^ghi^

^1^ Different small letters (a–n) indicate a significant difference (*p* < 0.05) among the yeast strains for the indicated metabolite concentration in each column.

**Table 3 foods-12-00830-t003:** Correlation matrix of sugar consumption, CO_2_ production and ethanol, glycerol, succinic acid, and acetic acid concentrations in sweet dough (14% sucrose *w*/*w* dm flour) with 23 different yeast strains, resulting from performing the multivariate correlation test with correlation probabilities *p* < 0.001 for all values.

	Sugar Consumption	CO_2_	Ethanol	Glycerol	Succinic Acid	Acetic Acid
Sugar consumption	1.00	0.74	0.71	0.61	0.44	0.42
CO_2_	0.74	1.00	0.95	0.94	0.80	0.81
Ethanol	0.71	0.95	1.00	0.92	0.74	0.72
Glycerol	0.61	0.94	0.92	1.00	0.82	0.86
Succinic acid	0.44	0.80	0.73	0.82	1.00	0.83
Acetic acid	0.42	0.81	0.72	0.86	0.83	1.00

**Table 4 foods-12-00830-t004:** A total of 51 identified volatile compounds and their odor threshold value and odor analyzed with GC-MS in dough fermented with 23 different yeast strains.

Chemical Group	Compound	Odor Threshold (µg/kg) *	Odor **
Aldehydes	2-Methyl-1-propanal	0.35	Unknown
	2-Methyl butanal	1–13	Malty
	3-Methyl butanal	0.1–17	Malty
	*E*-2-Butenal		Pungent, suffocating
	Hexanal	0.28–25.5	Green, grassy
	Heptanal	0.18–3	Fatty
	*E*-2-Heptenal	13	Green, fatty
	Nonanal	0.08–0.19	Soapy, citrus
	*E*-2-Octenal	3	Fatty, perfume-like
	Benzaldehyde	350	Bitter, almond-like
	*E*-2-Nonenal	0.08–0.19	Green, cucumber-like
Esters	Ethyl acetate	870–32,600	Solvent-like, fruity
	Ethyl propanoate	7–19,000	Sweet, fruity, grape, pineapple
	Ethyl isobutyrate	0.022–57.5	Sweet, ethereal, fruity, alcoholic
	Ethyl 3-methylbutanoate	0.013–6.9	Fruity, sweet, apple, pineapple
	Isoamyl acetate	30–94	Sweet, banana, fruity
	Isobutyl 2-methylbutanoate		Fruity, citrus, melon, ethereal
	Ethyl hexanoate	1–55.3	Fruity
	Isopentyl 2-methylbutanoate		Sweet, creamy, fruity, apple, winey, cherry, raisin, berry
	2-Methylbutyl 2-methylbutanoate		Unknown
	Ethyl octanoate	5–3150	Fruity, sweet
Acids	Acetic acid	22,000–2,000,000	Pungent, sour, fruit
	Isobutyric acid	230	Sour, cheesy, dairy, creamy
	4-Hydroxybutanoic acid		Unknown
	Butanoic acid	173–6800	Rancid, butter-like
	2-Methylbutanoic acid	12–1580	Cheese, sweat, rancid
Alcohols	1-Propanol	6600	Pungent, fruity, fermented
	Isobutanol	38–40,000	Malty
	1-Butanol	500–1240	Malty, fruity, banana, oily
	1-Penten-3-ol	250	Green, vegetable, fruity
	Isopentanol	250	Malty
	Oxitol	100	Unknown
	1-Pentanol		Fermented, bready, cereal, fruity, ethereal
	2-Methyl-1-pentanol		Fruity
	2-Heptanol	6	Fruity, green, earthy, bitter, citrus-like
	1-Hexanol	5370–8000	Green, flowery
	2-Methyl-3-hexanol		Ethereal, fruity
	3-Ethoxy-1-propanol		Fruity
	Cyclohexanol	1	Camphoreous, mentholic, fruity
	1-Octen-3-ol	3–4.8	Mushroom, earthy, fungal, green, oily, vegetable, umami
	1-Heptanol	0.9–50	Solvent, fermented, oily, nutty, fatty, green
	1-Nonanol	2.7	Waxy, citrus, orange, oily, fatty, spicy
	1-Octanol	1100	Waxy, orange, rose
Ketones	3-Pentanone		Ethereal, acetone, fruity
	2,3-Butanedione	0.05–6.5	Sweet, buttery, creamy, milky
	2-Heptanone	6.8	Soapy
	3-Octanone		Mushroom, cheesy, moldy, fruity
	3-Hydroxy-2-butanone	259–150,000	Buttery, carrot-like
	Acetol		Sweet, green, burned
	6-Methyl-5- hepten-2-one	50	Green, vegetable, musty, apple, banana, green bean
Other	γ-Nonalactone	21–30	Sweet, fruity

* Odor threshold values were taken from [8,12,33,34]. ** Odor descriptions specified in bread were taken from [35]; if not specified in bread, matching odor descriptions were taken from [35,36].

## Data Availability

Data is contained within the article.

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
