# Peer review of "Study of the Fermentation Characteristics of Non-Conventional Yeast Strains in Sweet Dough"

_foods, 2023, doi:10.3390/foods12040830_

Round 1
Reviewer 1 Report
This paper studies the potential of unconventional yeast strains in sweet dough, which provides a theoretical basis for improving the utilization of yeast resources in bread baking, and is an article of great practical significance. However, there are several issues that need improvement:
1. The detection conditions of ion exclusion high performance liquid chromatography can be described in detail. (lines 133 to 134)
2. It is recommended to supplement the volatiles of unfermented dough as a control.
3. There is a contradiction in the conclusion. It states in the lines 308-311 that invertase activity affects sucrose consumption, while in lines 324-327 it states that invertase has no effect on sugar consumption.
Reviewer 2 Report
This manuscript provides information about the fermentation characteristics of non-conventional yeast strains from different food industries in sweet dough. The analytical work described in this paper was very well approached with the methodology used in the characterization of the above-mentioned process, and the interpretation of the data is sound. Since there is still incomplete knowledge about the usage of different yeast strains in bakery products, this type of study is justified. The paper is well-written, but some changes in revising their manuscript are advisable:
Abstract/ Conclusion: Please compare the given CO2 production range to the results in Table 2.
Table 1. Technical error “Torulaspora delb.rückii”
Lines 217-225: Because the significance difference is not shown in Table 2, did the authors statistically analyze the obtained results? Furthermore, the table shows that the T58 sample has similar values for the measured parameters to the SD sample.
Lines 361- 370: Moreover, the baking process should also be taken into account. You already mentioned that latter in the manuscript, but as you state in this paragraph the reasons for the profile of volatile compounds between the dough and the finished product, it would be useful to add that part here as well.
Figure 4: Please indicate more clearly in the figure caption what differences in relative concentrations mean.
